# ADAPTIVE ENVIRONMENTAL MODELING FOR TASK-ORIENTED LANGUAGE AGENTS

## ABSTRACT

Recent advancements in the realm of intelligent agents, particularly those employing large language models, have been notably significant. Notwithstanding these advancements, intelligent agents encounter substantial challenges, predominantly in interactive and dynamic scenarios such as online shopping, attributed to an absence of integrated environmental modeling. In this paper, we propose a task-oriented environmental adaptation approach, allowing language agents to autonomously model new environments. This approach comprises two pivotal phases: Pre-Task Environment Exploration and In-Task Environment Update. The Pre-Task Environment Exploration phase incorporates a greedy exploration strategy, leveraging an agent in the role of an Evaluator to optimally explore environmental information based on present observations and feasible actions. This strategy is implemented through a recursive algorithm, enabling agents to choose and execute the top-k scored actions, thereby efficiently forming an Action-Observation Tree as the initial environmental modeling. During the In-Task Environment Update phase, agents employ environmental information to enhance task performance. The information generated from task execution and interaction trajectories is used to refine environmental modeling. These processes are iteratively executed, achieving mutual enhancement. We conduct a systematic evaluation of the environmental modeling, assessing both its effectiveness and comprehensiveness. The results demonstrate that under our approach, agents can indeed construct accurate environmental modeling. Simultaneously, we observe a significant enhancement in agent performance on both the ALFWorld-Eco and the WebShop benchmark datasets due to the application of environmental modeling.

## 1 INTRODUCTION

In the evolving landscape of Natural Language Processing (NLP), Large Language Models (LLMs) (Chowdhery et al., 2022; Brown et al., 2020) have surfaced as prominent entities demonstrating unparalleled proficiency and accomplishing state-of-the-art performance across diverse domains. By utilizing extensive pre-training data and advanced training methodologies, these models have excelled in generating coherent and contextually relevant outputs. However, inherent limitations still exist in LLMs, primarily due to their constrained capacity to store information in fixed weights and the complexities arising from limited context. To mitigate these limitations, researchers have proposed a series of enhancements focusing on refining the models' capabilities in task-specific scenarios, introducing sophisticated reasoning and planning algorithms (Wei et al., 2022; Yao et al., 2023; Besta et al., 2023), integrating various feedback mechanisms (Ouyang et al., 2022; Madaan et al., 2023; Yao et al., 2022b; Shinn et al., 2023; Wang et al., 2023), and incorporating external tools (Schick et al., 2023; Shen et al., 2023; Qin et al., 2023).

However, when addressing a myriad of real-world scenarios—particularly interactive decision-making scenarios such as online shopping, where each action induces modifications in the web page—achieving efficient planning and rational operation by the language model remains elusive in the absence of integrated environmental modeling. Artificial environmental modeling may be a choice; however, external environments are continually fluctuating. For instance, a multitude of new webpages emerge daily, and apps frequently undergo updates, adding various new features and so forth. Consequently, this approach evidently does not seem to be a suitable option for widespread

adoption. Thus, a valuable question arises: *Can language agents be empowered to autonomously and adaptively model new environments*?

In this paper, we advocate for a task-oriented environmental adaptation approach, enabling language agents to independently model new environment. This method encapsulates two crucial phases: *Pre-Task Environment Exploration* and *In-Task Environment Update*. The Pre-Task Environment Exploration employs a greedy exploration strategy, utilizing an agent as an Evaluator to optimally explore environmental information grounded on current observations and viable actions. The Evaluator, exploiting the profound knowledge amassed during the pre-training phase of language models, exhibits versatility, providing more flexibility and sample-efficiency across diverse complex environments. This methodology is executed via a recursive algorithm, deploying agents to select and undertake top-k scored actions, thereby proficiently establishing an initial environmental representation. During the In-Task Environment Update phase, the initial representation obtained from environmental modeling can significantly enhance the agent's proficiency in addressing specific tasks. Concurrently, the trajectories formulated throughout the task resolution process are leveraged to incessantly refine the agent's environmental representation. This incorporates freshly uncovered paths, hence, amassing experiences from varied environmental explorations to assist in executing forthcoming tasks. Such iterative execution and subsequent refinement culminate in a more sophisticated comprehension of the environment, elevating the likelihood of successfully resolving ensuing queries.

To carry out experimental evaluations, we have established a new benchmark, ALFWorld-Eco, which is based on the original ALFWorld dataset. In this refined benchmark, agents are facilitated to complete 60 varied tasks. In our experiments, we scrutinize the quality of the environmental information acquired by the agent through exploration from two dimensions: effectiveness and comprehensiveness. Both metrics affirm the viability of our method. Simultaneously, we analyze the improvement in task performance attributed to our methodology. The experimental outcomes on both the ALFWorld-Eco and the WebShop benchmark datasets reveal that the explicit incorporation of environmental modeling into the context results in significant augmentation of the agent's performance.

The contributions of our study are threefold:

1. We underscores the necessity for integrated environmental modeling for LLMs in dynamic and interactive scenarios, and we pose a new question: Can language agents be empowered to autonomously and adaptively model new environments?

2. We introduce an environmental adaptation approach, enabling language agents to independently model new environment, which encapsulates two crucial phases: Pre-Task Environment Exploration and In-Task Environment Update.

3. We systematically evaluate the effectiveness and comprehensiveness of the environmental modeling. Concurrently, we find that environmental modeling can significantly enhance agent performance on both the ALFWorld-Eco and the WebShop benchmark datasets.

## 2 RELATED WORK

### 2.1 LANGUAGE AGENT

The field of research concerning intelligent agents based on language models is experiencing rapid advancements. This encompasses a spectrum from the classical enhancement of reasoning capabilities like the Chain of Thought (Wei et al., 2022) to the representative tool-utilizing approaches such as AutoGPT and HuggingGPT (Shen et al., 2023) in planning and solving paradigm. Researchers have taken a step further by exploiting the feedback and assessment capabilities inherent in language models, introducing innovations like the Tree of Thought (Yao et al., 2023) and Graph of Thoughts (Besta et al., 2023), aiming to solve increasingly complex problems. The issues addressed by these methodologies, such as solving mathematical problems or planning how to schedule APIs given specific tools and problems, inherently contain comprehensive information, so language models can leverage their intrinsic knowledge to achieve global planning. However, in more realistic scenarios, such as online shopping on websites (Yao et al., 2022a) or benchmarks like TextWorld (Shridhar et al., 2021), every action incurs varying changes in the environment, necessitating models to make sequential decisions. React (Yao et al., 2022b) incorporates environmental feedback to support rea-

soning, and Reflection (Shinn et al., 2023) combines internal and external feedback. The method we propose assists these agents in explicitly modeling their environment, thereby enabling them to adapt to their surroundings and make more reasonable plans and decisions.

## 2.2 ENVIRONMENTAL MODELING AND EXPLORATION BY REINFORCEMENT LEARNING

In the domain of Reinforcement Learning (RL), environmental modeling stands as a pivotal component, aiding agents in learning and decision-making processes. A widely-adopted approach to environmental modeling involves leveraging Markov Decision Processes (MDP) to formalize environments, as portrayed by Sutton & Barto (2018). Delving into deep learning, neural network models emerge as quintessential tools for approximating environmental dynamics. Studies by Mnih et al. (2015) explicate the integration of Convolutional Neural Networks (CNN) to approximate environmental states in RL. The works by Ha & Schmidhuber (2018) serve as exemplary instances of employing world models, allowing agents to predict future states and formulate optimal policies, thereby enhancing their efficacy across various tasks. Contrary to these classical reinforcement learning algorithms, in our approach, intelligent agents based on language models utilize tree-structured textual forms to represent the environment.

In the realm of Reinforcement Learning (RL), the exploration strategies employed by agents are crucial in determining their ability to understand and interact effectively with their environments. One of the foundational strategies for agent exploration is the Epsilon-Greedy method, a probabilistic approach to explore-exploit dilemmas, illustrated by Sutton & Barto (2018). Furthermore, studies like Pathak et al. (2017); Burda et al. (2018a) highlight the incorporation of curiosity-driven learning models, where agents are rewarded for exploring states that are uncertain or less predictable, fostering a deeper and more nuanced interaction with diverse environmental aspects. Burda et al. (2018b) learns to predict the target network's activations, which encourages exploration by rewarding the agent for encountering states and actions that it cannot predict accurately. Ostrovski et al. (2017) combines count-based exploration with sparse rewards environment which helps the agent explore states that are rarely encountered, as these states yield higher bonuses. Unlike these exploration methods, considering the limited computational resources and to maximize the utilization of the inherent knowledge and capabilities of the language models, we adopt a greedy exploration approach before the task. During the task, we dynamically update the representation of the environment.

## 3 METHOD

We propose a specific method for environmental adaptation, as depicted in Figure 1, which involves dynamically and explicitly modeling the environment as a tree structure within the context. Utilizing a tree structure is inherently suitable for interactive decision-making environments, where the nodes represent environmental observations and the edges represent actions. Different actions lead to different observations. The exploration of the environment is conceptualized as a process of expanding nodes and edges of a tree. Given an environment and a set of optional actions, our method allows the agent to autonomously build and update its environmental modeling, specifically including two main phases: (1) Pre-Task Environment Exploration and (2) In-Task Environment Update.

## 3.1 PRE-TASK ENVIRONMENT EXPLORATION

Considering the vast exploration space found in many real-world scenarios and the limitations in the contextual length of language models, it is imperative for us to explore the most crucial environmental information to assist us in performing downstream tasks. Consequently, we adopt a greedy exploration strategy aimed at maximizing the exploration of environmental information. Given the current observation $O$ of the environment and a set of available actions $A$, this strategy employs an Evaluator to identify actions $(a_i, i = 1, 2, \ldots, k)$ that optimize information gains concerning the environmental representation.

$$a_i = \arg\max_{a \in A} \text{Evaluator}(O, a) \quad \text{for } i = 1, 2, \ldots, k \tag{1}$$

In our methodology, considering the diversity and complexity of environments, we propose employing the agent itself to evaluate states. When applicable, such a refined heuristic could be more

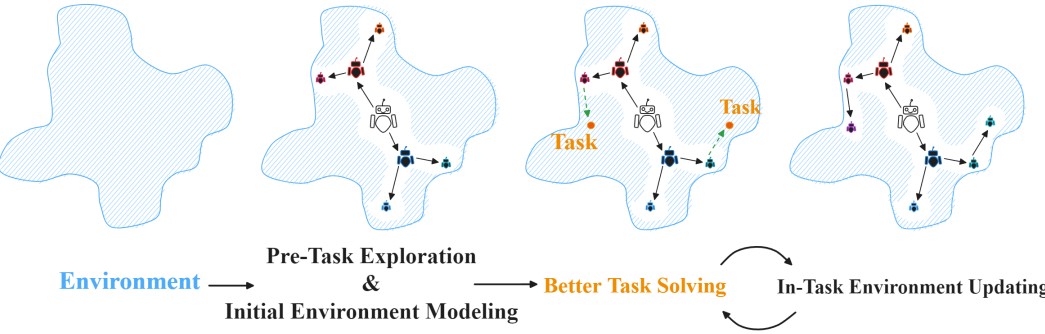

Figure 1: Illustration of Our Method. During the Pre-Task Environment Exploration, the agent constructs an initial environmental model by executing actions to gather observations, aiming to maximize the exploration of environmental information. In the task-solving process, the intelligent agent utilizes the environmental modeling to aid in task resolution. Concurrently, the trajectories generated from interactions with the environment during task resolution are employed to further optimize and update the environmental modeling. This interaction facilitates mutual enhancement between the two, subsequently elevating the agent's environmental modeling and adaptation capabilities.

flexible than hard-coded rules and more sample-efficient than learned models. Importantly, the extensive knowledge acquired by language models during pre-training not only endows the Evaluator with a relatively reliable basis for decision-making but also renders this method potentially universally applicable across various environments. During the specific evaluation process, the Evaluator may opt to score each action individually, e.g., from 1 to 10, and then select the top-k actions, or Evaluator could input all candidates concurrently, allowing the language model to directly pick the top-k actions.

We implement the exploration process using a recursive algorithm. When the agent selects the top-k actions, we dispatch agents with identical historical context information to execute the top-k actions respectively. The complete algorithm is illustrated in Algorithm 1. Through this algorithm, we obtain the initial $Tree_0$.

## 3.2 IN-TASK ENVIRONMENT UPDATE

As the agent engages in various tasks such as online shopping, it will handle some queries $Q$. The agent integrates the previously explored environmental information, $Tree_t$, and available actions, $A$, to interact with the environment $E$ in attempts to address the queries. Simultaneously, it receives new observation, represented as $O$, from the environment and generates trajectories $T$.

$$T = \text{Interact}(\text{Agent}(Q, Tree_t, A), E) \qquad (2)$$

Prior methodologies, including react and reflection methods, have strived to enhance task performance by conducting meticulous analysis of observation $O$ and trajectories $T$. Different from them, our methodology not only improves task performance but also continually refines its environmental representation by leveraging trajectories:

$$Tree_{t+1} = \text{Update}(Tree_t, T) \qquad (3)$$

Within a specified environment, multiple queries are commonly resolved. We iteratively execute Equations 2 and 3. Each execution of a query results in an environmental update. The resolution of more queries leads to a heightened understanding of the environment. Enhanced comprehension of the environment, in turn, elevates the probability of resolving subsequent queries.

In practical applications, these two phases can either be interpreted within a unified framework or as distinct methodologies. The efficacy of the task performance is directly proportional to the extent of the pre-task environment exploration; however, this implies a reduced benefit from the in-task environment update, and vice versa. In real-world scenarios, if the objective is to ensure

---

**Algorithm 1:** Explore Environment and Build Initial Action-Observation Tree

---

**Input:** Agent $\mathbb{A}$, Environment $\mathbb{E}$ and Exploration Depth $D$
**Output:** Action-Observation Tree $Tree_0$
**init** $Tree_0$
$\mathbb{T}$.root.observation $\leftarrow$ GetObservation($\mathbb{E}$)
$ActionsHistory \leftarrow []$

**procedure** NODEEXPLORE(node, $d$, $actions_{history}$)
**if** $d = 0$ **then**
$\quad | \quad$ **return**
**else**
$\quad$ $actions_{available} \leftarrow$ GetActionSpace(node.observation)
$\quad$ $actions_{next} \leftarrow \mathbb{A}$.SelectActions(node.observation, $actions_{available}$)
$\quad$ **for** *each $act_{next}$ in $actions_{next}$* **do**
$\quad\quad$ Reset($\mathbb{E}$)
$\quad\quad$ **for** *each $act_{history}$ in $actions_{history}$* **do**
$\quad\quad\quad | \quad$ $\mathbb{A}$.Execute($\mathbb{E}$, $act_{history}$)
$\quad\quad$ $\mathbb{A}$.Execute($\mathbb{E}$, $act_{next}$)

$\quad\quad$ $O \leftarrow$ GetObservation($\mathbb{E}$)
$\quad\quad$ NODEEXPLORE(node.AddChild($act_{next}$, $O$), $d - 1$, $actions_{history} + [act_{next}]$)

**return**
**end procedure**

NODEEXPLORE($Tree_0$.root, $D$, $ActionsHistory$)
**return** $Tree_0$

---

optimum and stable performance of the intelligent agent, consideration should be given to allocating additional computational resources for exploration. If resources are constrained, emphasis can be placed on in-task environment update to acquire environmental information through the trajectory of task execution, necessitating minimal additional resources.

## 4 EXPERIMENT

### 4.1 AGENT SETTING

We integrate the widely embraced React framework, serving as the agent, to evaluate the efficiency of environmental modeling meticulously. This framework is deployed for both the preliminary environmental exploration and executing downstream tasks. It is noteworthy that the original paper adopted the text-davinci-003 as the core language model; however, as OpenAI will soon cease to offer this API, we opted for more prevalent language models. Consequently, the core language models available for the agent are gpt-3.5-turbo and gpt-4, respectively. While implementing, we strive to align maximally with React's original methodologies: we explicitly define the methods for reason and action within the prompt. Additionally, we meticulously disclose every parameter associated with API calls. The temperature is precisely calibrated to zero; the output is strictly confined to a maximum token limit of one hundred; the top-p value is firmly anchored at one, and both frequency and presence penalties are unequivocally nullified.

### 4.2 BENCHMARK

WebShop. Yao et al. (2022a) proposed a sophisticated web-based problem-solving benchmark Web-Shop designed to assess agents' ability to adeptly navigate through an e-commerce website, with the objective to accurately locate and secure products in response to client requests. This environment is enriched with a diverse array of 1.18M real-world products accompanied by 12k articulate human instructions. WebShop encapsulates an extensive variety of both structured and unstructured

texts, including product titles, descriptions, and diverse options meticulously crawled from Amazon. It mandates agents to execute purchases of products, strictly aligning with user instructions—for instance, "I am looking for a nightstand with drawers. It should have a nickel finish, and be priced lower than $140," necessitating intricate web interactions, such as conducting searches for "nightstand drawers" and making selections like "color: modern-nickel-white" or opting to "back to search." The evaluation of this intricate task is orchestrated through an average score, representing the percentage of desired attributes covered by the selected product across all episodes. Our evaluations are conducted meticulously on the first 100 distinct test instructions, ensuring comprehensive assessment and validation of the agents' proficiency in managing complex, real-world e-commerce navigations and transactions.

ALFWorld-Eco. Shridhar et al. (2021) creat the ALFWorld to align with the embodied ALFRED benchmark (Shridhar et al., 2020), is a sophisticated, synthetic text-based game designed to challenge agents to navigate and perform multi-step tasks within a range of interactive environments. It incorporates six diverse types of tasks, each requiring the agent to accomplish a high-level goal, like examining a paper under a desk lamp, by navigating and interacting with a simulated household through textual actions (e.g., go to coffee table 1, take paper 2, use desk lamp 1). The original ALFWorld dataset is constructed with a task-centric focus, containing 134 different tasks in the unseen dataset, with nearly every task corresponding to a distinct environment. This approach does not align with our objective, as we aspire to build a new dataset centered around environments, allowing the execution of multiple tasks within a single environment. Consequently, we have developed ALFWorld-Eco, a new environment based on ALFWorld, where agents can complete 60 different tasks, categorized into six types, with each category comprising ten specific tasks. We employ the success rate as our metric for evaluation. The agent is assigned a score of 1 upon the successful completion of the task, and conversely, it receives a score of 0 if it fails to complete the task. We will release this new benchmark alongside the corresponding code. Our ALFWorld-Eco focuses on fostering a more diversified interaction within each environment, enabling agents to achieve a wider range of tasks and goals. This shift from task-centric to environment-centric design empowers agents to better understand and adapt to varying contextual circumstances within the same environment, enhancing their flexibility and applicability in real-world scenarios.

## 4.3 EVALUATION

The experiments are primarily concerned with two aspects. On one hand, we necessitate the evaluation of the quality of the environmental information procured by the intelligent agent, utilizing our proposed methodology. On the other hand, it is imperative to evaluate the enhancement in task performance attributed to the dynamically updated environmental information. In the experiment, for the pre-task environment exploration, we default to the settings of $k = 3$ and $n = 3$ for both benchmark environments. Addressing the requirements of more advanced intelligent agents in the future and considering real-world application scenarios, our intelligent agents should autonomously adapt to entirely new environments without relying on human intervention, such as data annotation. Therefore, we opt for a zero-shot evaluation rather than a few-shot evaluation. This setup is employed for both environment exploration and downstream tasks.When addressing queries within WebShop, we configure the agent with a maximum step limit of 15; for ALFWorld-Eco, we set the limit to 40 steps. If the agent fails to resolve the queries within the allocated steps, it is deemed unsuccessful.

### 4.3.1 EVALUATING ACQUIRED ENVIRONMENTAL INFORMATION

We assess the quality of the environmental information acquired by the intelligent agent through exploration from two perspectives: effectiveness and comprehensiveness.

*Effectiveness:* This criterion fundamentally evaluates the effectiveness of acquiring pertinent environmental information. If the actions undertaken by the intelligent agent are executable, then the information fed back from the environment is considered effective. Conversely, if actions are non-executable or fail to comply with certain norms or specifications, leading to a lack of environmental feedback, they are deemed ineffective. Our observations suggest that, when the instructions explicitly require the intelligent agent to choose from available actions, agents developed with both GPT-3.5 and GPT-4 frameworks succeed in generating nearly 100% executable actions within both environments, rendering the returned environmental information consistently effective. We princi-

pally attribute this elevated level of precision to the reduction of judgment difficulty provided by the recursive architecture in our method, coupled with the inherent effectiveness of instruction tuning in large language models.

Table 1: Comprehensiveness Assessment of Environmental Information

| LLM Core | Stage | Webshop | ALFWorld-Eco |
|----------|-------|---------|--------------|
| GPT-3.5 | Pre-task | 68.2 | 12.7 |
| | In-task | 85.8 | 65.4 |
| GPT-4 | Pre-task | 77.4 | 16.2 |
| | In-task | 93.1 | 81.3 |

*Comprehensiveness:* We further evaluated the comprehensiveness of the information, initially obtaining a gold-truth global tree description of the entire environment by traversing all actions. Subsequently, we needed to assess the amount of information at each node. One approach is to directly consider the quantity of text each node contains; however, despite some nodes containing extensive text, they don't necessarily reflect environmental information. For instance, elaborate descriptions of specific products do not substantially aid in understanding the Webshop environment. Thus, we adopted a manual evaluation method, specifically utilizing a crowdsourcing approach, allowing ten individuals familiar with the relevant environment to score the amount of information contained in the environmental observations corresponding to each action, with a total score of 100. The average score for each node serves as the numerical representation of the environmental information it holds. The cumulative value of each node in the tree represents the overall environmental information content. We use the total information content of the tree acquired by the intelligent agent divided by the total information content of the global tree as a score to measure its comprehensiveness.

The results listed in the table indicate that exploration at different stages is beneficial for acquiring environmental information. For example, in the Webshop environment, under our method and parameter settings, a considerable amount of information has already been explored during the pre-task phase. Conversely, in the case of ALFWorld-Eco, owing to its expansive exploration space, the portion of information unearthed during the initial stages is comparatively diminutive on a global scale. However, following exploration during the in-task phase, the agent realizes substantial information gains post-task. Concurrently, GPT-4 exhibits enhanced exploratory proficiencies relative to GPT-3.5.

*Analysis of Case Study:* The discrepancies observed in the results during the pre-task environment exploration phase primarily originate from Equation 1. These discrepancies are due to the fact that agents, when based on different language models, may opt for varied actions even with the same observation. Additionally, agents with varying language models demonstrate different levels of proficiency in extracting vital insights about their environments. Appendix Figures 3 and 4 represent the outcomes of in-task environment exploration within a Webshop scenario, utilizing agents anchored in GPT-3.5 and GPT-4 models, respectively. A divergence is witnessed at $n = 3$; the top three actions elected by the GPT-3.5-based agent were [Buy Now], [Description], and [Prev], contrasting with [Description], [Features], and [Attributes] chosen by the GPT-4-based agent. Despite differences in initial modeling, agents from both GPT-3.5 and GPT-4 were able to develop comprehensive environment models as the subsequent in-task environment exploration advanced. A comparison between Figures 4 and 5 uncovers additional enhancements made in environment modeling throughout the in-task Environment Exploration. Intriguingly, the agents were proficient in identifying anomalies within the environment. For instance, in the Webshop scenario, selecting [Next] on the search results page resulted in a direct error, stating "Invalid action!" A thorough examination revealed that a page limitation was instituted in the original environment setting in React, allowing the environment to render only the first page of the search results. This subtle environmental anomaly echoes real-world scenarios more closely, leading to our decision to retain this configuration.

### 4.3.2 ENVIRONMENTAL ADAPTATION FOR TASK ENHANCEMENT

This section primarily demonstrates the enhancement in task performance attributed to the dynamically updated environmental information. The adaptive method ensures that the agents are equipped with some preliminary environmental information before they resolve the query, optimizing their

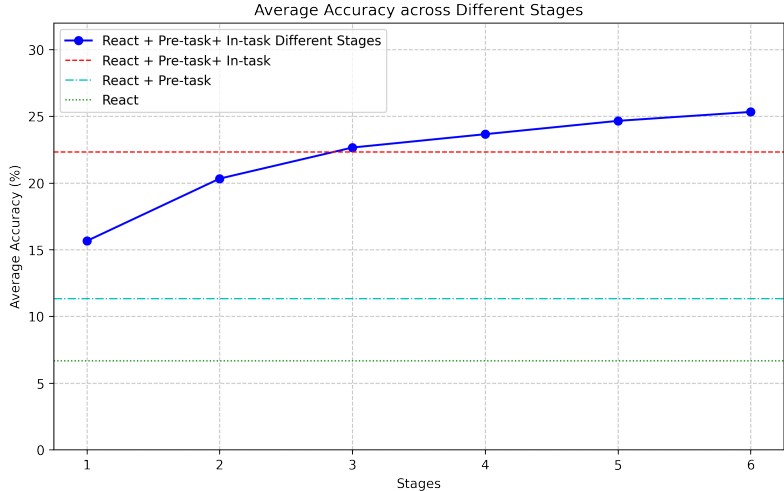

Figure 2: Average Accuracy across Different Stages on ALFWorld-Eco

performance in the tasks by continuously updating their environmental awareness (Equation 2 and Equation 3).

Table 2: Scores of React and React+ Environmental Adaptation in Different Environments

| Environments | Webshop | ALFWorld-Eco |
|---|---|---|
| React | 14.95 | 6.67 |
| React+ Environmental Adaptation | 24.25 | 18.33 |

As depicted in Table 2, introducing the environmental adaptation method significantly improved the average performance of the agents, which feature a GPT-3.5-turbo as their language model core. We further conducted a bad case analysis on React and React+ Adaptation, identifying three types of errors related to environmental interaction information:

1. Executing the search action directly on the product page instead of returning to the search page;

2. Encountering an error due to the agent selecting the [Next] button in the environment;

3. Continuously browsing products without clicking 'buy now' until reaching the maximum number of steps.

Agents equipped with environmental adaptation largely avoided the aforementioned errors. The main cases where they failed were when they reached the maximum number of steps during multiple refinements of search keywords for products.

We delve even further into analyzing the advantages yielded by environmental adaptation during the in-task phase within the ALFWorld-Eco environment. Specifically, 60 tasks are randomly assorted, into six distinct groups, each encompassing 10 tasks. The average accuracy is then calculated within each group, representing the accuracy for varying stages during the in-task phase. To mitigate the potential bias introduced by varying difficulties of different tasks, we conduct a total of five rounds of repeated experiments. Subsequently, the average accuracy of the six groups at diverse stages is once more averaged per round to denote the final accuracy. As shown in Figure 2, there is a noticeable increase in accuracy in the later stages, illustrating that, during the in-task phase, the agent's continuous adaptation to the environment systematically amplifies its capability to resolve tasks. The figure presents the ablation performance of the method, where "React" indicates the exclusive application of the React method without any environmental adaptation. "React+Pre-task" denotes that the examination only occurs in the pre-task phase, with no updates occurring in the in-task phase. A comparison of these results reveals that both exploration and updating are crucial

for the agent's performance in the ALFWorld-Eco environment, underscoring the importance of the two components.

## 4.4 DISCUSSION

Considering the inherent limitations in contextual length, when the environment is rife with complexities and the trajectories are proliferating, the expansion of the tree becomes inevitable, necessitating critical truncations. Although we have not encountered such a situation in this paper, we have also attempted to engage in a discussion about it. Specifically, we will maintain and dynamically update the frequency of each node appearing in the trajectories $T$ throughout the task execution process. If the initial frequency values of all nodes are 0, and they appear on the correct trajectories, their frequencies are increased by 1. When a reduction in environmental representation is crucial, nodes with lower frequencies are prioritized for deletion. This mechanism enables task-driven dynamic environmental adaptation, allowing the agent to seamlessly adjust to varying task demands and environmental complexities.

## 5 CONCLUSION

Our approach strives to address the imperative needs of LLMs in real-world dynamic scenarios, emphasizing autonomous environmental modeling to augment their decision-making processes both before and during the task execution, proving pivotal in enhancing language agent effectiveness and adaptability in varied applications. Moving forward, we aim to undertake evaluations employing a wider array of agents and more genuine datasets. Additionally, we hope to explore how, upon integration of environmental modeling, various global planning methodologies can be amalgamated to further elevate the performance of sequential decision-making tasks that necessitate interaction with the environment.

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

# A APPENDIX

## A.1 ACTION-OBSERVATION TREE EXAMPLE

```
gpt-3.5 pre-task:

Webshop Environment
└─ Instruction
    ├─ Requirement: Specific type and specification of product
    ├─ Condition: Price below a certain value
    │
    └─ Action: Search
        └─ Search Query: Keywords based on given requirements and conditions
            │
            └─ Observation: Search Results
                ├─ [Back to Search]
                │   └─ Action: Click
                │       │
                │       └─ Observation: Returned to Initial Blank Search Page
                └─ [Result Page 1]
                    ├─ [Result 1]
                    │   ├─ Name: Product Name
                    │   ├─ Details: Product Details
                    │   ├─ Price: Product Price
                    │   │
                    │   └─ Action: Click
                    │       │
                    │       └─ Observation: Product Page
                    │           ├─ [Back to Search]
                    │           ├─ [< Prev]
                    │           │   └─ Action: Click
                    │           │       │
                    │           │       └─ Observation: Previous Product Page
                    │           ├─ Available Types: Multiple selectable types
                    │           ├─ Available Specifications: Multiple selectable specifications
                    │           ├─ Price: Price for the selected specification
                    │           ├─ Rating: Rating for this product
                    │           │
                    │           ├─ [Description]
                    │           │   └─ Action: Click
                    │           │       │
                    │           │       └─ Observation: Detailed Product Description
                    │           ├─ [Features]
                    │           ├─ [Reviews]
                    │           ├─ [Attributes]
                    │           └─ [Buy Now]
                    │               └─ Action: Click
                    │                   │
                    │                   └─ Observation: Your score
                    ├─ [Result 2]
                    │   ├─ Name: Product Name
                    │   ├─ Details: Product Details
                    │   └─ Price: Product Price
                    ├─ [Result 3]
                    │   ├─ Name: Product Name
                    │   ├─ Details: Product Details
                    │   └─ Price: Product Price
                    │
                    ├─ [Back to Search]
                    │   └─ Action: Click
                    │       │
                    │       └─ Observation: Returned to Initial Blank Search Page
                    │
                    └─ [Next >]
                        └─ Action: Click
                            │
                            └─ Observation: Invalid action!
```

Figure 3: GPT-3.5 Pre-task Action-Observation Tree Structure.

```
gpt-4 pre-task:

Webshop Environment
└─ Instruction
    ├─ Requirement: Specific type and specification of product
    ├─ Condition: Price below a certain value
    │
    └─ Action: Search
        ├─ Search Query: Keywords based on given requirements and conditions
        │
        └─ Observation: Search Results
            ├─ [Back to Search]
            │   └─ Action: Click
            │       │
            │       └─ Observation: Returned to Initial Blank Search Page
            └─ [Result Page 1]
                ├─ [Result 1]
                │   ├─ Name: Product Name
                │   ├─ Details: Product Details
                │   ├─ Price: Product Price
                │   │
                │   └─ Action: Click
                │       │
                │       └─ Observation: Product Page
                │           ├─ [Back to Search]
                │           ├─ [< Prev]
                │           ├─ Available Types: Multiple selectable types
                │           ├─ Available Specifications: Multiple selectable specifications
                │           ├─ Price: Price for the selected specification
                │           ├─ Rating: Rating for this product
                │           │
                │           ├─ [Description]
                │           │   └─ Action: Click
                │           │       │
                │           │       └─ Observation: Detailed Product Description
                │           ├─ [Features]
                │           │   └─ Action: Click
                │           │       │
                │           │       └─ Observation: Detailed Product Features
                │           ├─ [Reviews]
                │           │   └─ Action: Click
                │           │       │
                │           │       └─ Observation: User Reviews and Ratings
                │           ├─ [Attributes]
                │           │
                │           └─ [Buy Now]
                ├─ [Result 2]
                │   ├─ Name: Product Name
                │   ├─ Details: Product Details
                │   └─ Price: Product Price
                ├─ [Result 3]
                │   ├─ Name: Product Name
                │   ├─ Details: Product Details
                │   └─ Price: Product Price
                ├─ [Back to Search]
                │   └─ Action: Click
                │       │
                │       └─ Observation: Returned to Initial Blank Search Page
                └─ [Next >]
                    └─ Action: Click
                        │
                        └─ Observation: Invalid action!
```

Figure 4: GPT-4 Pre-task Action-Observation Tree Structure.

```
gpt-4 +in-task:
Webshop Environment

└─ Instruction
    ├─ Requirement: Specific type and specification of product
    ├─ Condition: Price below a certain value
    │
    └─ Action: Search
        └─ Search Query: Keywords based on given requirements and conditions
            │
            └─ Observation: Search Results
                ├─ [Back to Search]
                │   └─ Action: Click
                │       │
                │       └─ Observation: Returned to Initial Blank Search Page
                └─ [Result Page 1]
                    ├─ [Result 1]
                    │   ├─ Name: Product Name
                    │   ├─ Details: Product Details
                    │   ├─ Price: Product Price
                    │   │
                    │   └─ Action: Click
                    │       │
                    │       └─ Observation: Product Page
                    │           ├─ [Back to Search]
                    │           │   └─ Action: Click
                    │           │       │
                    │           │       └─ Observation: Returned to Initial Blank Search Page
                    │           ├─ [< Prev]
                    │           │   └─ Action: Click
                    │           │       │
                    │           │       └─ Observation: Previous Product Page
                    │           ├─ Available Types: Multiple selectable types
                    │           ├─ Available Specifications: Multiple selectable specifications
                    │           ├─ Price: Price for the selected specification
                    │           ├─ Rating: Rating for this product
                    │           ├─ [Description]
                    │           │   └─ Action: Click
                    │           │       │
                    │           │       └─ Observation: Detailed Product Description
                    │           ├─ [Features]
                    │           │   └─ Action: Click
                    │           │       │
                    │           │       └─ Observation: Detailed Product Features
                    │           ├─ [Reviews]
                    │           │   └─ Action: Click
                    │           │       │
                    │           │       └─ Observation: User Reviews and Ratings
                    │           ├─ [Attributes]
                    │           │   └─ Action: Click
                    │           │       │
                    │           │       └─ Observation: Detailed Product Attributes
                    │           │
                    │           └─ [Buy Now]
                    │               └─ Action: Click
                    │                   │
                    │                   └─ Observation: Your score
                    ├─ [Result 2]
                    │   ├─ Name: Product Name
                    │   ├─ Details: Product Details
                    │   └─ Price: Product Price
                    ├─ [Result 3]
                    │   ├─ Name: Product Name
                    │   ├─ Details: Product Details
                    │   └─ Price: Product Price
                    │
                    ├─[Back to Search]
                    │   └─ Action: Click
                    │       │
                    │       └─ Observation: Returned to Initial Blank Search Page
                    │
                    └─ [Next >]
                        └─ Action: Click
                            │
                            └─ Observation: Invalid action!
```

Figure 5: GPT-4 Pre-task + In-task Action-Observation Tree Structure.