# OpenReview forum: "Adaptive Environmental Modeling for Task-Oriented Language Agents"
_ICLR.cc/2024/Conference — Submitted to ICLR 2024_

### Official Review · Reviewer_aS2x · 2023-10-21

**Soundness:** 2 fair
**Presentation:** 3 good
**Contribution:** 2 fair
**Rating:** 3
**Confidence:** 4

**Summary:**

The authors propose a task-oriented environmental adaptation approach to model new environment with two phases: Pre-Task Environment Exploration and In-Task Environment Update. The authors conduct experiments on ALFWorld-Eco and WebShop benchmark datasets to validate the proposed method.

**Strengths:**

originality

The authors attempt to improve task-oriented language agent with environment model. It would be a novel approach; however, there are issues with the design: both the underlying LLM and the "environment model" from Pre-Task Environment Exploration and In-Task Environment Update are approximations, which will cause problems to agent's performance.

quality

The environment model as the authors build is not the same as defined in standard AI textbooks. The performance metric "score" is not clearly defined.

clarity

There are some issues in the writing as discussed above.

significance

There are issues in the paper as discussed above.

**Weaknesses:**

Both the underlying LLM and the "environment model" are approximations, which will cause problems to agent's performance.


The performance metric "score" is not clearly defined and the method is not clear and not objective.

**Questions:**

A.

3.1 PRE-TASK ENVIRONMENT EXPLORATION


"we propose employing the agent itself to evaluate states"
Such self-reference method is fundamentally flawed since the underlying LLM is not perfect, and there is no study about how different this may deviate from a perfect model.

B.

How to score is not clearly defined.

C.

Algorithm 1
It is for deterministic environments only, i.e., it does not represent uncertainties in action takings / transitions.

D.

When we talk about environment model, we talk about state/observation transition model. Algorithm 1 samples the environment model to build a tree. It is over-claiming to call it modeling the environment.


E.

4.3.1 EVALUATING ACQUIRED ENVIRONMENTAL INFORMATION
Comprehensiveness
"we adopted a manual evaluation method, specifically utilizing a crowdsourcing approach, allowing ten individuals familiar with the relevant environment to score the amount of information contained in the environmental observations corresponding to each action, with a total score of 100"

The method appears very vague, subjective and arbitrary.

---

> ### Author Response · Authors · 2023-11-23
>
> Thank you very much for your detailed feedback. We suspect that you might be a reviewer with a background in reinforcement learning. Some differences in background might have created a significant gap between what we intended to express and what you understood.
> Firstly, we would like to offer some further elaboration and clarification regarding our motivation, and then we will address your concerns in detail.
>
> ### **Clarification about Motivation and Contribution**
>
> We deeply regret that our assumptions about the reader's background may have led to an underemphasis on the motivation of our work, causing some misunderstandings. We hope that the subsequent comparison table and explanatory text will provide you with a clearer understanding of the underlying motivation behind our study.
>
> | Agent | Feedback | Modeling | Adaptive |
> | --- | --- | --- | --- |
> | React[1] | ✓ |  |  |
> | Reflexcion[2] | ✓ |  |  |
> | Autogpt[3] | ✓ |  |  |
> | Tree of thought[4] | ✓ |  |  |
> | Graph of thought[5] | ✓ |  |  |
> | Ours | ✓ | ✓ | ✓ |
>
> Clearly, environmental information is greatly beneficial for agents in solving tasks. Previously, agents based on large language models primarily utilized environmental feedback for decision-making. Our approach goes a step further by explicitly incorporating environmental modeling into the context, which is the core motivation of our work.  As we continue to consider how to model environments using text, we find that external environments are continually fluctuating, therefore it is best to let the LLM-based agent adaptively model.
>
>  Starting from a perspective of resource and contextual limitations, we propose a relatively practical and adaptive method. This involves using the language model itself to explore the environment and employing textual descriptions for environmental modeling. Undoubtedly, the agent concepts in reinforcement learning are more mature than those in LLM-based methods, the design philosophy of this method is partly inspired by reinforcement learning such as greedy exploration. Our experiments demonstrate that explicitly introducing environmental modeling into the context indeed enhances the model's performance.
>
> [1][Synergizing Reasoning and Acting in Language Models](https://arxiv.org/abs/2210.03629)
>
> [2][Reflexion: Language Agents with Verbal Reinforcement Learning](https://arxiv.org/abs/2303.11366)
>
> [3]https://github.com/Significant-Gravitas/AutoGPT
>
> [4][Deliberate Problem Solving with Large Language Models](https://arxiv.org/abs/2305.10601)
>
> [5][Graph of Thoughts: Solving Elaborate Problems with Large Language Models](https://arxiv.org/abs/2308.09687)
>
> Below we provide responses to the concerns and questions you have raised:
>
> ### **Reply to questions A,C,D ：Environment Model / Environment Modeling**
>
> We sincerely appreciate your insightful review from the perspective of classical reinforcement learning, which has brought to our attention the potential for confusion in our terminological expressions. In agreement with your perspective, we acknowledge that the term "environment model" indeed possesses a very standard and established definition within the realm of reinforcement learning. Recognizing this concern, as you might have observed, we have carefully avoided using the term "environment model" to describe the static Action-Observation Tree in our work. This deliberate choice stems from our understanding that to align with the conventional definition of an environment model, one would need to integrate additional components, like the state/observation transition model. Our approach has been to employ the concept of "environment modeling" solely to describe the objective and process. In this process, we determined that the Action-Observation Tree we developed was sufficiently comprehensive for the language model's environmental understanding, thereby eliminating the necessity to construct the other aspects typically associated with a standard environment model.

---

> ### Author Response · Authors · 2023-11-23
>
> ### **Reply to questions A**
>
> **1.The self-reference method seems fundamentally flawed, given that the underlying Large Language Model (LLM) is not perfect.**
>
> It's important to highlight that Algorithm 1 is responsible for building the Action-Observation Tree, with the LLM being just a component of this process. We have taken into account your concern about the LLM's imperfections. The design of Algorithm 1 greatly lessens the dependency on the LLM's performance, thereby reducing the risk of potential flaws, such as hallucinations, influencing the outcomes. The role of the language model is simply to choose the most exploratory action from a given set of observations and executable actions. Thus, the language model is more than capable of performing its designated role within the algorithm. This is evidenced in our experiments, as described in Section 4.3.1, where it is noted that “agents developed with both GPT-3.5 and GPT-4 frameworks succeed in generating nearly 100% executable actions within both environments”.
>
> **2.There's a lack of study on how this might differ from a perfect model.**
>
> As detailed in Section 4.3.1 of our experiments, the language model consistently produces 100% executable actions when presented with observations and a set of executable actions. This indicates that the Action-Observation Trees constructed by Algorithm 1 are inherently accurate. The primary aspect of their imperfection is not incorrectness, but incompleteness. The Comprehensiveness score in Section 4.3.1 assesses how far these trees are from being complete Action-Observation Trees.
>
> ### **Reply to questions B,E：Comprehensiveness Score**
>
> For specific tasks like Webshop and ALFWorld-Eco, the scores are typically based on conventional accuracy or , in other words, success rates. Your query seems to center around the Score for Comprehensiveness. Let’s explore this in detail: Given your expertise in reinforcement learning, you can think of this Score as akin to a weighted State Coverage. The 'weight' in this context is indicative of the informational value of each action in enriching our understanding of the environment from a human perspective. Take the Webshop example: for the observation [result page 1], which includes [result 1]  (a), [result 2] (b), [result 3] (c), [Back to Search] (d), and [Next >] (e). The letters in "( )" representing the potential informational value of each clickable action. Assuming resources allow the agent to explore two actions, an agent 1 opts to explore [result 1] and [Next >], its score would be calculated as $\frac{a + e}{a + b + c + d + e} $. Conversely, if another agent 2 explores [result 1] and [result 2], its score would be $\frac{a + b}{a + b + c + d + e} $.
>
> In this context, for question E, absent a human-centric evaluation and assuming equal weights for all actions, both agents 1 and 2 would end up with the same score of $\frac{2}{5} $, thereby diminishing the significance of the comprehensiveness evaluation. To address the issue of potential human bias, we have meticulously developed a set of Scoring Criteria for Comprehensiveness.
> - **Information Relevance**: Evaluate how relevant the information at each node is to the overall understanding of the Webshop environment. Irrelevant or overly detailed information about specific products should be scored lower.
> - **Depth of Environmental Insight**: Score based on how deeply the node's information contributes to understanding the environment. Nodes providing unique or essential insights should receive higher scores.
> - **Clarity and Conciseness**: Assess whether the information is clear and to the point. Unnecessarily verbose or unclear descriptions should be scored lower.
>
> We also incorporate assessments from multiple crowd-sourced evaluators to ensure a balanced perspective, while paying close attention to the consistency of the scoring system.
>
> **Regarding consistency**: To assess score distribution consistency, we first conducted an intuitive check and found that the score distribution was relatively consistent. We then proceeded with a quantitative statistical analysis. To minimize the influence of individual crowd workers' scoring preferences, we normalized the scores for each worker. This involved dividing each node's score by the total node scores and then multiplying by 100. Subsequently, for each node, we calculated the standard deviation of scores given by different crowd workers. Finally, we computed the average of these standard deviations across all nodes. The results were 0.76 for Webshop and 0.42 for ALFWorld-Eco, indicating a relative consistency in the crowd workers' scoring.
>
> Thank you once again for your constructive feedback. If our responses have effectively addressed your concerns, we would deeply appreciate it if you could re-evaluate our work and grace us with a positive score. If you have any new suggestions, we will also incorporate them into the revised version.

---

### Official Review · Reviewer_7M58 · 2023-10-31

**Soundness:** 3 good
**Presentation:** 1 poor
**Contribution:** 2 fair
**Rating:** 5
**Confidence:** 3

**Summary:**

The paper presents a task-oriented environmental adaptation approach for intelligent agents, particularly those using large language models, to autonomously model new environments in dynamic scenarios like online shopping. The approach is divided into two phases: Pre-Task Environment Exploration, which uses a greedy exploration strategy to form an initial environmental model, and In-Task Environment Update, which refines this model based on task execution and interaction data, leading to improved task performance.

I have several major comments and questions:

- In the abstract it’s not clear what is the role of language agents, and ‘language’ in particular in the proposed solution/innovation. All described is the process of recursive greedy exploration and update, a process that is not novel and has been around for some years now.

- Similar issue with the introduction. I still don’t understand what’s the role of language or language agents in the recursive refinement process. The introduction needs to be significantly revised to focus on the central innovation and contribution proposed by this paper, which I believe is around the use of language and language agents, rather than the recursive environment model update, which is not novel.

- It is not very clear to me, from the introduction, what is the critical problem that this approach is trying to solve. The paper needs to be motivated much better. Try providing intuitive examples that why adaptive environment modeling is a significant problem that needs to be addressed. You may also consider adding a mathematically rigorous problem formulation section.

- The first point mentioned in the list of contributions in not a contribution. It more sounds like a motivating question. Please remove. It’s better to only have two contributions than overselling a motivating point as a significant contribution to the field.

- Please apply the revisions suggested in my first two comments to the second contribution item as well. Again, I believe the contribution is around the use of language and language agents for environment modeling, rather than the recursive environment model update. The latter is not novel.

- I read through the methodology, and unfortunately, still don’t have a solid understanding about the role of ‘language’ in the proposed solution. Is language used for the refinement process, for instance by posing critical questions regarding the greedy actions made and their outcome? This is unclear and need to be explained in more details.

- If I understand correctly, during the refinement process, the history of the greedy actions is considered and then the environment model is updated. Is that correct? But what is the update process, specifically, as described in equations 2-3. Also, is there a mechanism to refine the greedy action policy? Is the greedy action policy adopted for the entire model refinement process? How to be sure that the greedy action execution policy is the best approach throughout the model refinement process?

- The evaluations and results are significantly insufficient. More test cases, benchmarks and metrics are needed for a solid conclusion. In the only result figure presented, why are the accuracies so low? Why aren’t any of the ablations learning anything? Such questions need a detailed discussion. Also, error bars must be plotted to see if the differences are statistically significant. This also applies to results and numbers reported in the tables. Overall, the evaluation section is very weak and needs significant improvements.

- What are the limitations of the approach. The limitations are never discussed.

At current states I vote for rejection mostly due to weak motivation, unclear presentation, unclear contributions, and weak evaluation. I need to see more discussions and revisions as suggested above, as well as suggested by my fellow reviewers to make this a ICLR-ready paper. I’d be happy to increase my score when authors satisfactorily addressed the comments.

**Strengths:**

See above.

**Weaknesses:**

See above.

**Questions:**

See above.

---

> ### Author Response · Authors · 2023-11-23
>
> Thank you very much for your detailed feedback. We suspect that you might be a reviewer with a background in reinforcement learning. Some differences in background might have created a significant gap between what we intended to express and what you understood. In this regard, we hope to further discuss this with you. Below we provide responses to the concerns and questions you have raised:
>
> ### **Reply to comments and questions 1, 2, 3, 4, 5: Unclear motivation**
>
> We deeply regret that our assumptions about the reader's background may have led to an underemphasis on the motivation of our work, causing some misunderstandings. Please understand that concepts like 'recursive greedy exploration and update' or 'using language agents' are not the primary drivers of our research, but rather elements of our proposed solution.
>
> We hope that the subsequent comparison table and explanatory text will provide you with a clearer understanding of the underlying motivation behind our study.
>
> | Agent | Feedback | Modeling | Adaptive |
> | --- | --- | --- | --- |
> | React[1] | ✓ |  |  |
> | Reflexcion[2] | ✓ |  |  |
> | Autogpt[3] | ✓ |  |  |
> | Tree of thought[4] | ✓ |  |  |
> | Graph of thought[5] | ✓ |  |  |
> | Ours | ✓ | ✓ | ✓ |
>
> Clearly, environmental information is greatly beneficial for agents in solving tasks. Previously, agents based on large language models primarily utilized environmental feedback for decision-making. Our approach goes a step further by explicitly incorporating environmental modeling into the context, which is the core motivation of our work.  As we continue to consider how to model environments using text, we find that external environments are continually fluctuating, therefore it is best to let the LLM-based agent adaptively model.
>
>  Starting from a perspective of resource and contextual limitations, we propose a relatively practical and adaptive method. This involves using the language model itself to explore the environment and employing textual descriptions for environmental modeling. Undoubtedly, the agent concepts in reinforcement learning are more mature than those in LLM-based methods, the design philosophy of this method is partly inspired by reinforcement learning such as greedy exploration. Our experiments demonstrate that explicitly introducing environmental modeling into the context indeed enhances the model's performance.
>
> We will follow your advice to update the abstract and introduction accordingly.
>
> [1][Synergizing Reasoning and Acting in Language Models](https://arxiv.org/abs/2210.03629)
>
> [2][Reflexion: Language Agents with Verbal Reinforcement Learning](https://arxiv.org/abs/2303.11366)
>
> [3]https://github.com/Significant-Gravitas/AutoGPT
>
> [4][Deliberate Problem Solving with Large Language Models](https://arxiv.org/abs/2305.10601)
>
> [5][Graph of Thoughts: Solving Elaborate Problems with Large Language Models](https://arxiv.org/abs/2308.09687)
>
> ### **Reply to comments and questions 6,7：The use of  greedy action policy and language.**
>
> **Greedy Action Policy**
>
> During the initial stage of our environmental exploration, we deliberately employ a greedy action policy, enabling the model to choose from the top-k actions. This strategy is designed to maximize the model’s exploration of the environment by drawing upon the inherent knowledge of the LLM-based agent. However, in the second phase, our focus shifts to task resolution. We move away from a greedy approach in task execution. This is because the integration of environmental modeling is intended to facilitate a more globally informed action selection process. In the 'Think' phase of React, the agent can apply the environmental modeling insights for comprehensive reasoning and planning, leading to the selection of actions that are more suitable.
>
> **the Use of Language Model**
>
> The language model is primarily used as an Evaluator in the initial exploratory phase  and Task Executor in the second stage.
>
> **1.Evaluator**: As illustrated in Algorithm 1, in the step of actions_next ← A.SelectActions(node.observation, actions_available), given the observation and the available actions set, the action that maximizes the exploration of the environment is selected by the language model. The specific prompt is essentially as follows:
>
> ```
> Activity: You are an agent engaged in exploring the Webshop site.
>
> Objective: Maximize environmental exploration by selecting appropriate actions, through iteratively performing the Think and Action steps outlined below:
>
> Think: Given the observation [ the observation details ], consider which actions would maximize the acquisition of environmental information.
>
> Action: Based on your formulated strategies, choose the top-k appropriate actions to interact with the environment.
>
> The actions you take must be one of the following:
> "think[strategy]" to accomplish the Objective.
> "search[product]" to look for specific products.
> "click[button]" to interact with actionable elements. Only strings in "[ ]" are buttons.
>
> ```

---

> ### Author Response · Authors · 2023-11-23
>
> **2.Task Executor**:
>
> ```
> Activity: You are an agent and engaged in an online shopping task on the Webshop site.
>
> Environment：Below are the interaction for the Webshop : [Action-Observation Tree](Similar to figure 3,4,5 in appendix)
>
> Objective: Complete the purchasing task concerning 'the instruction' by iteratively performing the Think and Action steps outlined below:
>
> Think: Given the observation [ the observation details ], think how you can accomplish the instruction task.
>
> Action: Based on your formulated strategies, perform an action to interact with the environment.
>
> The actions you take must be one of the following:
> "think[strategy]" to accomplish the instruction task.
> "search[product]" to look for specific products.
> "click[button]" to interact with actionable elements. Only strings in "[ ]" are button.
>
> ```
>
> Then you can set the instruction task.
>
> **3. Updating the Environment Tree with New Trajectories.**
>
> As the Agent carries out its tasks, it concurrently constructs an Action-Observation Tree, essentially forming new trajectories. Thus, the update as defined in Formula 3 fundamentally represents a fusion of two distinct trees. Notably, this phase proceeds without the  language model. We meticulously engage in a breadth-first traversal of the nodes within the trajectory-representing tree. Whenever we encounter nodes absent in the tree that depicts the environment, we refine and expand the latter by integrating these new nodes. To illustrate, refer to the comparative analysis of the Action-Observation Trees in figures 4 and 5 in the appendix. Figure 5 vividly demonstrates the enrichment of the tree with novel nodes, such as [Buy Now] and [Attributes], introduced through the progression of task trajectories.
>
> ### **Reply to comments and questions 8：Evaluations and Results**
>
> **1.More Test Cases**
>
> In the webshop dataset, we initially used only 100 instruction tasks. When we expanded it to 300 tasks, the agent's effectiveness (success rate) continued to show a slight increase.
> | Environments                | Webshop   |
> |----------------------------|---------|
> | (100) React                        | 14.95   |
> | (100) React + Environmental Adaptation | 24.25   |
> | (300) React + Environmental Adaptation | 26.13   |
>
>
>
> **2.More Benchmarks**
>
> More benchmarks are certainly beneficial. To our knowledge, most suitable LLM-based agent benchmarks are still in the process of being refined and their corresponding papers are under review. We are unsure about the research community's acceptance of the quality of these LLM-based agent benchmarks. If you have any mature benchmarks to recommend, we would be more than willing to further refine our work. In fact, the new environment ALFWorld-Eco that we constructed in our paper is also aimed at enabling more comprehensive benchmark assessments.
>
> **3.More metrics**
>
>  Beyond success rate, we are currently unsure what other comprehensive and reasonable metrics might be. Could you specify any particular metrics? If you mean error bars and statistical errors, we assure you that these will be included in future updated versions.
>
> **4.The reason for the lower results**
>
> The reason for the lower results is that our approach is completely zero-shot, without any examples. Introducing a few-shot learning could indeed significantly improve performance, but it does not align with the problem scenario we are considering. We want language agents to adapt autonomously to new environments, especially those unexplored by humans, and thus we do not include few-shot examples with annotations and detailed task and environment instructions."
>
> **5.Abaltions**
>
>  Section 4.3.2 and Figure 2 include the Ablation Studies. It seems you have missed them: The figure presents the ablation performance of the method, where “React” indicates the
> exclusive application of the React method without any environmental adaptation. “React+Pre-task” denotes that the examination only occurs in the pre-task phase, with no updates occurring in the in-task phase. A comparison of these results reveals that both exploration and updating are crucial for the agent’s performance in the ALFWorld-Eco environment, underscoring the importance of the two components.
>
> **6. Statistically Significant and Error Bars**
>
> As detailed in section 4.1,  to ensure consistent results, we meticulously calibrated the temperature to zero. This precision calibration leads to remarkably consistent performance of the language model across several experiments on the same dataset, demonstrating minimal variability. Additionally, to reinforce the statistical robustness of Figure 2, we implemented a strategy of randomizing the dataset order and conducting repeated averages. Although it's not possible to present images here, rest assured that Error Bars will be prominently featured in the final version of the document .

---

> ### Author Response · Authors · 2023-11-23
>
> **7.Limitations**
>
> The primary aim of our approach is to model the environment within a context. However, real-world environments can be complex and may not be comprehensively described within the textual limitations of context length. Two developments could alleviate this potential limitation. First, as mentioned in the discussion section of the paper, is the introduction of a deletion mechanism. Second, research on expanding the context length of LLMs is advancing rapidly, and there are already some LLMs available that handle extended contexts.
>
> Thank you once again for your constructive feedback. If our responses have effectively addressed your concerns, we would be immensely grateful if you could provide us with a positive score. If you have any new suggestions, we will also incorporate them into the new version.

---

### Official Review · Reviewer_KgBF · 2023-10-31

**Soundness:** 2 fair
**Presentation:** 2 fair
**Contribution:** 2 fair
**Rating:** 5
**Confidence:** 3

**Summary:**

This paper proposes a novel approach to enable language agents to autonomously model new environments, thereby enhancing their task performance. The approach involves two phases: Pre-Task Environment Exploration and In-Task Environment Update. The authors evaluate the approach using two environments: Webshop and ALFWorld-Eco. The results show that the proposed method is beneficial for modeling environmental to solve task better.

**Strengths:**

1. The approach enables language agents to autonomously model new environments, thereby enhancing their task performance.
2. The authors evaluate the approach using two environments and provide detailed results.

**Weaknesses:**

1. Many details in the method were not explained clearly. In the first stage of environmental exploration, the language model is used as a Evaluator for validation. Please provide the corresponding prompt. In the second stage of environment update, no details were provided on how to interactively generate new trajectories and how to update the environment tree using the new trajectories, such as the corresponding prompt.
2. There were also many details that were not clearly written in the experiment. In a Comprehensiveness evaluation, the criteria for manual scoring and the consistency of manual scoring should be given. Is the accuracy used in Table 2 and Figure 2 in ALFWorld Eco. Please provide details on what different stages refer to in Figure 2.
Because many details are not explained clearly, it is also difficult to judge the contribution of the work.

**Questions:**

N/A

---

> ### Author Response · Authors · 2023-11-23
>
> Thank you for taking the time to review this paper and provide detailed feedback. We sincerely appreciate the insightful comments and the critical examination of our work. Below we provide responses to the concerns and questions you have raised:
>
> ### **Reply to Weaknesses1: Provide details about the usage of the language model and the prompt.**
>
> The language model is primarily used as an Evaluator in the initial exploratory phase  and Task Executor in the second stage.
>
> **1.Evaluator**: As illustrated in Algorithm 1, in the step of actions_next ← A.SelectActions(node.observation, actions_available), given the observation and the available actions set, the action that maximizes the exploration of the environment is selected by the language model. The specific prompt is essentially as follows:
> ```
> Activity: You are an agent engaged in exploring the Webshop site.
>
> Objective: Maximize environmental exploration by selecting appropriate actions, through iteratively performing the Think and Action steps outlined below:
>
> Think: Given the observation [ the observation details ], consider which actions would maximize the acquisition of environmental information.
>
> Action: Based on your formulated strategies, choose the top-k appropriate actions to interact with the environment.
>
> The actions you take must be one of the following:
> "think[strategy]" to accomplish the Objective.
> "search[product]" to look for specific products.
> "click[button]" to interact with actionable elements. Only strings in "[ ]" are buttons.
> ```
> **2.Task Executor**:
> ```
> Activity: You are an agent and engaged in an online shopping task on the Webshop site.
>
> Environment：Below are the interaction for the Webshop : [Action-Observation Tree](Similar to figure 3,4,5 in appendix)
>
> Objective: Complete the purchasing task concerning 'the instruction' by iteratively performing the Think and Action steps outlined below:
>
> Think: Given the observation [ the observation details ], think how you can accomplish the instruction task.
>
> Action: Based on your formulated strategies, perform an action to interact with the environment.
>
> The actions you take must be one of the following:
> "think[strategy]" to accomplish the instruction task.
> "search[product]" to look for specific products.
> "click[button]" to interact with actionable elements. Only strings in "[ ]" are button.
> ```
> Then you can set the instruction task.
>
> **3. Updating the Environment Tree with New Trajectories.**
>
> As the Agent carries out its tasks, it concurrently constructs an Action-Observation Tree, essentially forming new trajectories. Thus, the update as defined in Formula 3 fundamentally represents a fusion of two distinct trees. Notably, this phase proceeds without the  language model. We meticulously engage in a breadth-first traversal of the nodes within the trajectory-representing tree. Whenever we encounter nodes absent in the tree that depicts the environment, we refine and expand the latter by integrating these new nodes. To illustrate, refer to the comparative analysis of the Action-Observation Trees in figures 4 and 5 in the appendix. Figure 5 vividly demonstrates the enrichment of the tree with novel nodes, such as [Buy Now] and [Attributes], introduced through the progression of task trajectories.
>
>  ### **Reply to Weaknesses2 : Provide details about experiment**
>
> **1.Scoring Criteria for Comprehensiveness**
>
> - Information Relevance: Evaluate how relevant the information at each node is to the overall understanding of the Webshop environment. Irrelevant or overly detailed information about specific products should be scored lower.
> - Depth of Environmental Insight: Score based on how deeply the node's information contributes to understanding the environment. Nodes providing unique or essential insights should receive higher scores.
> - Clarity and Conciseness: Assess whether the information is clear and to the point. Unnecessarily verbose or unclear descriptions should be scored lower.
>
> **2.Regarding Consistency**
>
> To assess score distribution consistency, we conducted an  intuitive check and found that the score distribution was relatively consistent. We then proceeded with a quantitative statistical analysis. To minimize the influence of individual crowd workers' scoring preferences, we normalized the scores for each worker. This involved dividing each node's score by the total node scores and then multiplying by 100. Subsequently, for each node, we calculated the standard deviation of scores given by different crowd workers. Finally, we computed the average of these standard deviations across all nodes. The results were 0.76 for Webshop and 0.42 for ALFWorld-Eco, indicating a relative consistency in the crowd workers' scoring.
>
> **3.Is the accuracy used in Table 2 and Figure 2 in ALFWorld Eco ？**
>
> Yes，accuracy or, in other words, success rate.

---

> ### Author Response · Authors · 2023-11-23
>
> **4.Details on what different stages of Figure 2**
>
> In section 4.3.2, there are approximately six lines of text describing the details of different stages, starting from "Specifically, 60 tasks are randomly assorted into six distinct groups…" and ending with "the average accuracy of the six groups at diverse stages is once more averaged per round to denote the final accuracy." To simplify, the stages primarily emphasize the sequence of events. For example, stage 1 represents the first 10 tasks executed in that environment. As the agent performs more tasks, it becomes increasingly adapted to the environment, thus improving its success rate.
>
> Thank you once again for your constructive feedback. If our responses have been helpful in addressing your concerns, we would greatly appreciate a positive score. And we look forward to hearing from you.
>
> Best regards.

---

### Author Response · Authors · 2023-11-23
**General Response**

We are grateful for the meticulous reviews from multiple reviewers. Your responsible attitudes are deeply appreciated.

We have noted that the main concerns of the reviewers focus on three points:

1. Details of the use of an LLM-based agent, such as specific prompts.
2. Details on the Comprehensiveness score.
3. The motivation is not clear enough: there may be some background differences that could lead to a significant gap between what we intend to express and what the reviewers understand.

We believe these issues can be well clarified through a rebuttal. Please see the detailed responses to each reviewer. We also commit to updating these points in the next version of paper.

---

### Meta-Review · Area_Chair_iCRN · 2023-12-14

**Metareview:**

The paper proposes an approach for LLM agents to model new environments autonomously. The process consists of two phases: Pre-Task Environment Exploration and In-Task Environment Update. The evaluation is done on the ALFWorld-Eco and WebShop benchmark datasets.

Reviewer KgBF commented on the weaknesses in methodological clarity and missing experimental details. As a response, the authors provided detailed prompts used in the language model. Reviewer 7M58 also rated the paper marginally below the acceptance threshold, expressing concerns about the unclear role of language in agents, weak motivation, and insufficient evaluations. The author response had a positive ,but limited impact on addressing the reviewer's concerns regarding the novelty and clarity of the paper. KgBF increased the rating but still rated the paper as marginally below the acceptance. Reviewer aS2x recommended rejection, highlighting multiple issues with evaluation. The authors responded by clarifying their methodology and the rationale behind their scoring system. Despite these clarifications, the reviewer maintained their score, indicating that the responses did not sufficiently address the reliability of the proposed method and its evaluation pipelines.

After reviewing the comments and the authors' responses, significant concerns remain regarding the depth and clarity of the methodology and contributions, and the reliability of the evaluation process. AC recommends a reject decision. The authors are encouraged to address these concerns comprehensively in future revisions, possibly by including a broader range of environments to substantiate the efficacy of their proposed method.

**Justification For Why Not Higher Score:**

Overall the paper received consistently negative ratings from the reviewers. The paper would benefit from a thorough revision addressing the areas mentioned by the reviewers, particularly focusing on enhancing the clarity of the methodology, demonstrating the unique contributions more convincingly, and strengthening the reliability of its evaluation process. A few more notes regarding evaluation: While AC is receptive to the application of LLM in evaluation processes, as evidenced by successful instances in tasks like Visual Question Answering (VQA), the evaluation framework proposed in this paper is fundamentally different. This difference necessitates a more comprehensive justification to ascertain that the use of LLMs for evaluation in this specific context can indeed yield dependable and meaningful signals. Alternatively, the authors may consider enhancing their evaluation by incorporating additional metrics, which could involve some degree of manual effort.

**Justification For Why Not Lower Score:**

N/A

---

### Decision · Program_Chairs · 2024-01-16

Reject